# Multiple Strategies to Develop Small Molecular KRAS Directly Bound Inhibitors

**DOI:** 10.3390/molecules28083615

**Published:** 2023-04-21

**Authors:** Xile Zhou, Yang Ji, Jinming Zhou

**Affiliations:** 1Department of Colorectal Surgery, The First Affiliated Hospital, Zhejiang University School of Medicine, 79 Qingchun Road, Hangzhou 310003, China; 2Drug Development and Innovation Center, College of Chemistry and Life Sciences, Zhejiang Normal University, 688 Yingbin Road, Jinhua 321004, China

**Keywords:** mutated KRAS, drug discovery, KRAS inhibitors, directed binding

## Abstract

KRAS gene mutation is widespread in tumors and plays an important role in various malignancies. Targeting KRAS mutations is regarded as the “holy grail” of targeted cancer therapies. Recently, multiple strategies, including covalent binding strategy, targeted protein degradation strategy, targeting protein and protein interaction strategy, salt bridge strategy, and multivalent strategy, have been adopted to develop KRAS direct inhibitors for anti-cancer therapy. Various KRAS-directed inhibitors have been developed, including the FDA-approved drugs sotorasib and adagrasib, KRAS-G12D inhibitor MRTX1133, and KRAS-G12V inhibitor JAB-23000, etc. The different strategies greatly promote the development of KRAS inhibitors. Herein, the strategies are summarized, which would shed light on the drug discovery for both KRAS and other “undruggable” targets.

## 1. Introduction

KRAS (Kirsten rat sarcoma viral oncogene homolog) gene mutation is widespread in tumors and is one of the most frequently mutated oncogenes in various malignancies, including lung, colon, and pancreatic cancers, etc.; it is found in one out of every seven human cancers [1,2]. Typically, KRAS switches between the GTP-bound “on” state and the GDP-bound “off” state [3,4]. The activating mutations drive the KRAS protein in a GTP-bound active conformation, thereby resulting in a state of sustained activation of KRAS signaling, which in turn causes uncontrolled proliferation of the cells and leads to carcinogenesis [4,5]. Moreover, KRAS-activating mutations can inhibit the apoptosis of tumor cells, modulate cell metabolism, and alter the tumor microenvironment to induce tumor immunosuppression, thus further promoting the metastasis of the tumor [6,7]. Therefore, targeting KRAS mutations is regarded as the “holy grail” of targeted cancer therapies. However, targeting KRAS mutations directly has been a huge challenge since its discovery in 1982, when it was deemed the “undruggable target” [3,8].

Strikingly, due to the improved understanding of the structural biology of KRAS and the development of drug-designing technologies, covalent KRAS inhibitors that block mutant KRAS-G12C have recently been successfully developed. Sotorasib (FDA, 2022) and adagrasib (FDA, 2022) were approved, and exhibited immediate and profound clinical impacts in oncotherapy [9,10]. Furthermore, the inhibitors which target other KRAS-activating mutants including KRAS-G12D (MRTX1133) [11] or KRAS-G12V (JAB-23000) are in preclinical studies [12]. Interestingly, multiple developing drug design strategies have been applied to the KRAS inhibitor discovery, including targeted protein degradation (TPD) technology [13], multivalent binding strategy, and salt bridge strategy [14,15]. Therefore, the drug development targeting KRAS provides efficient therapeutic agents for KRAS mutation-driven tumors and sheds light on the drug discovery of undruggable targets. Herein, from the view of drug development, we focus on recent trends in KRAS inhibitors, including structures, mechanisms, and R&D strategies.

## 2. Structural Basis of KRAS Function

KRAS is one of the downstream genes of EGFR receptor activation; it is about 35 kb long, and is located on chromosome 12 [16]. The K-RAS gene can encode two types of protein variants, KRAS-4A and KRAS-4B proteins, of which the 4B variable spliceosome is the predominant one and is commonly referred to as the KRAS protein [17] (Figure 1A). The KRAS protein is in a class of membrane-bound GTPases, which is controlled by two classes of proteins, guanosine nucleotide exchange factors (GEFs) and GTPase-activating proteins (GAPs) through the interconversion between the active GTP-bound state and the inactive GDP-bound state [18,19]. Among them, GEFs facilitate the exchange of GTP with bound GDP to activate K-RAS, while GAPs promote the hydrolysis of GTP to GDP and inactivate KRAS proteins. Activated EGFR proteins promote the conversion of GDP to GTP in GDP-KRAS protein by recruiting the GRB2 protein to bind the GEF factor SOS (Son of Sevenless) [20,21]. Activated KRAS induces a series of downstream pathways, especially the rapidly accelerated fibrosarcoma (RAF)-mitogen-activated protein kinase kinase (MEK) extracellular regulated protein kinases (ERK), and PI3K-serine/threonine-protein kinase (AKT)-mammalian target of rapamycin (mTOR) pathways, thereby promoting cell proliferation, transcription, protein synthesis, survival, metabolism, and other life activities (Figure 1B) [22,23,24]. Specifically, mutations in KRAS disrupt the guanine exchange cycle, resulting in KRAS “locking” in the active GTP-bound state, thereby activating downstream signaling pathways, including RAF/MEK/ERK and PI3K/AKT/mTOR [25,26].

KRAS is composed of 3 domains, including the effector lobe (residues 1−86), the allosteric lobe (residues 87−166), and the hypervariable region (HVR) (residues 167−188) (Figure 1C) [27,28]. The effector lobe consists of the phosphate binding loop (P-loop) (residues 10−17), the switch I region (residues 30−38), and the switch II region (residues 60−76), which serves as a nucleotide-binding motif functioning as a GTP-GDP switch. For the GTP-bound on state, the γ-phosphate of GTP forms two hydrogen bonds with Thr35 (switch I) and Gly60 (switch II), respectively, which holds the two switch regions in an active conformation to facilitate the recruitment of effector proteins [29]. The hydrogen bonds are destroyed when GTP is hydrolyzed into GDP, which releases the switch regions and returns the GDP-bound inactive conformation. The allosteric lobe domain is involved in the membrane interactions and dimerization of KRAS, while the HVR is implicated in the recognition and association of KRAS with the membrane, which is essential for the recruitment of effector proteins [29]. The process of membrane binding is triggered by the prenylation of the cysteine at the HVR’s CAAX box (cysteine, 2 aliphatic amino acids, another residue) via farnesyltransferase (FTase). Subsequently, the AAX motif is cleaved by RAS converting enzyme 1 (RCE1), while cysteine is methyl-esterified by isoprenylcysteine carboxyl methyl transferase (ICMT), making KRAS protein rich in hydrophobic tails, which bear a high binding affinity with cell membrane [30]. The cell membrane localization of KRAS is required for its activity. Recent evidence indicates that the dimerization of KRAS monomers is important to their downstream signaling activity [31].

## 3. KRAS Mutations in Cancer

As the most frequently mutated oncogenes, KRAS-activated mutation is found in one out of every seven human cancers [1,2]. Various studies show that malignant activation of the KRAS oncogene was specifically associated with human carcinogenesis [32,33]. In addition to inducing cell proliferation, KRAS mutations modulate the tumor microenvironment (TME) and alter the infiltration of immune cells and the expression of the cytokine expression [6,7]. Mutant KRAS increases the expression of inflammatory cytokines such as CXCL-8, IL-1, and NF-κB [6,34], which promote tumorigenesis by elevating the vascularity and invasiveness of tumors, stromal remodeling, and immune suppression. The high prevalence of KRAS mutations has been identified to play an important role in the immunosuppressive TME in cancers, through actions such as inducing NLRP3 inflammasome activation and programmed death-ligand-1 (PD-L1) expression [35]. Chen et al. reported an association between KRAS mutations and the upregulation of PD-L1 via RAF/MEK/ERK pathway in lung adenocarcinoma (LAC) cell lines. On the other hand, the activation of the KRAS-downstream pathway PI3K/AKT/mTOR in LACs tightly modulates the expression of PD-L1 both in vitro and in vivo [36]. Furthermore, Liu et al. found KRAS mutations caused increased CD8+ tumor-infiltrating lymphocytes that were associated with tumor immunogenicity [37]. Overall, these results indicate the oncogenic KRAS can lead to immune escape by its downstream pathway via PD-L1.

KRAS mutations are found at different rates in a variety of cancers. KRAS mutations are most frequently found in pancreatic ductal adenocarcinoma (PDAC) (>85%), followed by colorectal cancer (CRC) (~40%), and non-small cell lung cancer (NSCLC) (~30%) [38,39]. The majority of KRAS mutations are single-base missense mutations, which are commonly identified at codons 12 (83%), 13 (14%), or 61 (2%) on exons 2 and 3 [25]. The mutations in these codons cause an increase in GTP binding affinity and the activation of KRAS to induce downstream signaling through different mechanisms. The G12C mutation induces a conformational change at the GAP binding site and dismisses GAP-mediated GTP hydrolysis. Moreover, oncogenic KRAS mutations in codons 12, 13, and 61 suppress GTP hydrolysis and keep KRAS in the activated state [25]. In addition to GAP-mediated hydrolysis, oncogenic KRAS mutations can also reduce rates of intrinsic GTP-hydrolysis by reducing the stability of the arginine residue hydrolysis transition state. A 40–80-fold decrease in intrinsic GTP-hydrolysis may be caused by G12A, G12R, Q61H, and Q61L mutations [25]. Recent molecular dynamics studies suggest that the G12D mutation alters local conformations and dynamics of KRAS [40,41]. Specifically, the distribution of KRAS mutations shows significant preferences in different human cancers. For instance, G12C mutations account for 41% of KRAS mutations in LUAD, whereas G12D and G12V are the two most common mutations in CRC and PDAC. The KRAS-G12R mutation, especially, is only identified in PDAC. Moreover, KRAS mutations often combine with specific co-mutations in TP53, STK11, or KEAP1, etc., which modulate the function of KRAS and lead to oncogenesis [38,42].

## 4. Challenge in Directly Targeting Oncogenic KRAS

The importance of oncogenic KRAS in tumorigenesis makes KRAS an extremely important target in oncotherapy. The proof of principle for targeting KRAS protein as an anti-cancer strategy has been verified by the knockout of KRAS protein in both human and mouse cell lines [43]. shRNA knockdown technology was used by Anurag et al. to study cell lines expressing KRAS protein, and found that these cell lines could be divided into two groups, KRAS-dependent and KRAS-independent. KRAS-dependent cell lines expressed E-cadherin factor, which is sensitive to shRNA knockdown of KRAS protein. Moreover, in transgenic mice, induction of KRAS-G12D protein expression using doxycycline resulted in tumor generation and promoted tumor growth, whereas cessation of induction of KRAS protein expression resulted in the shrinkage, or even clearing of the tumor [44].

Strategies for targeting mutant KRAS proteins include downregulation of KRAS expression using siRNA, inhibition of post-translational modification or downstream effectors of KRAS, and direct targeting of KRAS [9]. KRAS has been regarded as an undruggable target since its discovery, and designing inhibitors that directly bind KRAS protein was a huge challenge [45]. The difficulty in developing small molecule inhibitors targeting KRAS is closely related to the structural properties of KRAS protein. Despite how the binding pockets of GDP and GTP of KRAS are well suited as small molecule binding sites, GDP, GTP, and KRAS have binding affinities of a picomolar degree, and their concentrations in vivo are millimolar, making it impractical to develop compounds with an effective blockage. Moreover, in addition to nucleoside binding sites, there are a lack of binding pockets with sufficiently large and deep hydrophobic regions. For these reasons, small-molecule inhibitors that directly target KRAS have not been developed until recent years [46,47].

## 5. Strategies Directly Targeting Mutated KRAS

### 5.1. Covalent Binding Strategy

As more and more covalent drugs have been successfully used in the clinical setting, covalent binding strategies have attracted great interest in drug development [48,49]. For inhibitors targeting mutant KRAS, the covalent binding strategy represents a major breakthrough in drug development, which greatly improves the binding ability and selectivity of small molecular compounds for mutant KRAS proteins [50]. Such a strategy focuses on the KRAS-G12C mutation, and this class of compounds can specifically covalently bind to cysteine at previously unreported binding sites, rendering KRAS insensitive to nucleoside transforming factors such as SOS, thus leaving the conformation of KRAS-G12C in the GDP-bound inactivated state. Two compounds based on covalent binding strategies, sotorasib and adagrasib, have been approved by the FDA in 2022 [10,51,52]. Currently, a covalent binding strategy has been adopted in not only KRAS-G12C, but also KRAS-G12D and KRAS-G12S [15].

Cysteine is the most nucleophilic amino acid among the 20 common amino acids and is most commonly covalently bound in covalent drug design. The G12C mutation provides the possibility of covalent binding to the 12-position cysteine to enhance the binding affinity, thereby improving the inhibitory selectivity (Figure 2D). The first KRAS-G12C-specific inhibitors were developed based on the structure of a fragment of 6H05 in 2013 by Ostrem et al., which specifically bound in the allosteric site with the covalent linkage to C12 (Figure 2A); therefore, they were disrupting the recruitment of GTP and maintaining the inactive state [53]. Another type of inhibitor that binds to the allosteric site is the Quinazoline series, with the quinazoline ring as the core structure, which represents the most potent KRAS-G12C allosteric inhibitors, including MRTX849, AMG-510, ARS-1620, and AZD4625 [54,55,56,57] (Figure 2B). Moreover, KRAS-G12C allosteric inhibitors with other core structures have also been in development, including ARS-853, BI-0474, JDQ443, etc. (Figure 2C) [58,59,60].

Recently, due to the success of covalent kinase inhibitors, Lim et al. developed the first series of the covalent substrate (GTP/GDP)-competitive inhibitors targeting the catalytic site by modifying diphosphate compounds with various electrophiles and different linkers. SML-8-73-1 was identified as the lead compound, which could block >95% KRAS-G12C in competition tested in the presence of 1 mM GDP or GTP by irreversible binding and was regarded as a promising inhibitory agent for KRAS-G12C [61]. However, two negatively charged phosphate groups at SML-8-73-1 cause low membrane penetration, which limits further application in vivo. The optimization of SML-8-73-1 led to SML-10-70-1 (Figure 3A), which improved the cell permeability and competitively inhibited KRAS-G12C through covalent binding.

In addition to targeting KRAS-G12C, the covalent binding strategy has also been applied to KRAS-G12S and KRAS-G12D. Shakat’s group reported that a series of small molecules could suppress the oncogenic signaling of KRAS-G12S through the chemical acylation of a serine residue. The β-lactone group of design compounds such as G12Si-1 and G12Si-2 could acylate the mutant serine of KRAS-G12S to exhibit selectivity, making the compounds show activity in cells expressing KRAS-G12S, but sparing the wild-type KRAS (Figure 3B). This technology overcomes the weak nucleophilicity of an acquired serine residue, which may serve as a way to selectively target other inactivated serines in covalent drug development [62]. Interestingly, Goldsmith et al. developed a covalent inhibitor RMC-9805, for which the structure is not disclosed. RMC-9805 forms a non-covalent complex with KRAS-G12D and cyclophilin A first, and then the cool covalent warhead of RMC-9805 binds irreversibly with the mutant aspartic acid over a matter of minutes to hours [15].

The inhibitors covalently targeting oncogenetic mutant KRAS showed rather good anti-cancer activity. Treatment with AMG-510 caused the regression of KRAS-G12C tumors and enhanced the anti-tumor efficacy of either chemotherapy or targeted drugs. AMG-510 improved the pro-inflammatory tumor microenvironment and generated durable therapeutic effects alone or combined with immune-checkpoint inhibitors. Cured mice may have induced adaptive immunity against shared antigens to reject the growth of isogenic KRAS-G12D tumors [51]. Moreover, treated with another covalent inhibitor of KRAS-G12C ARS1620, the KRAS-G12C mutant cell would present ARS1620-modified peptides in MHC-I complexes, which could serve as tumor-specific neoantigens to elicit a cytotoxic T cell response against KRAS-G12C cells [63].

### 5.2. Targeted Protein Degradation Strategy

Currently, targeted protein degradation techniques such as proteolysis-targeting chimeras (PROTAC) have attracted great attention in drug development [64]; they are expected to be applied to undruggable targets. In 2020, Crews’s group developed the first PROTAC LC-2, which covalently bound KRAS-G12C with an MRTX849 warhead, and recruited the E3 ligase VHL to induce the degradation of KRAS-G12C (Figure 4) [65,66]. Moreover, Chen’s group developed the pomalidomide-based PROTAC degraders. Of them, compound KP-14 exhibited the highest KRAS-G12C degrading capability in NCI-H358 cells and showed potent antiproliferative activity (Figure 4) [67]. Following this, Lu’s group reported the first reversible covalent KRAS-G12C PROTAC YF135 based on a cyanacrylamide-based reversible covalent bond, which induced the rapid and sustained degradation of KRAS-G12C [68]. Zhang et al. developed a series of PROTACs based on AMG-510, and compound III-2 was identified to exhibit binding and degradation ability for KRAS-G12C, showing a more potent inhibitory effect on downstream p-ERK signaling (Figure 4) [69]. In addition to KRAS-G12C, Astellas developed a first-in-class degrader, ASP3082 (structure not shown), which efficiently degraded the G12D mutant and has been in clinical trials since June 2022 [15].

Beyond the heterodimeric small molecules such as PROTACs, the monomeric targeted protein degrader for KRAS is also reported, which induces the interaction between KRAS and E3 ligase to lead to the degradation of KRAS. A natural product, Kurarinone, which was reported to have anti-cancer activity against various cancers, was identified to decrease the protein level of KRAS by proteasomal degradation dependent on an E3 ubiquitin ligase WDR76. Knockdown of WDR76 through small interfering RNA siWDR76 restored the level of KRAS as well as the downstream protein p-ERK and c-MYC (Figure 4). Moreover, Kurarinone arrested the cell cycle in the G0/G1 phase in a p53-independent manner. However, the binding data for KRAS of Kurarinone are still not available [70]. Through the analysis of TOP flash reporter cells and potential toxicity effects on primary neural stem cells, Moon’s group identified CPD0857 from 2000 chemical compounds, which induced ubiquitin-dependent proteasomal degradation of Ras proteins. Moreover, CPD0857 effectively inhibited the proliferation and increased the apoptosis of CRC cell lines, and overcame the resistance of CRC harboring KRAS mutations. Accordingly, CPD0857 also inhibited tumor growth and significantly decreased Ras protein expression in xenograft tumors of mice [71].

### 5.3. Targeting the Dimerization of Oncogenic KRAS Strategy

The dimerization of oncogenic KRAS plays an important role in the activation of MAPK signaling to promote the proliferation of cancer cells. The salt bridge between D154 of one KRAS molecule and R161 of a partner is key for the formation of KRAS dimerization [72]. Marshall’s group identified that the α-β dimerization of KRAS-G12D is induced through the second phosphatidylserine-dependent interface, which is sensitive to small molecule inhibitors. Therefore, targeting the dimerization interface to inhibit KRAS–KRAS interaction has become a promising strategy [73]. It was reported that targeting the α4-α5 dimerization interface using RAS-specific monobody would inhibit oncogenic KRAS and inhibit tumor formation in vivo [74]. Specifically, the KRAS inhibitor BI-2852 was identified bound to a pocket between switch I and II on RAS with the nanomolar affinity, which is distinct from that of covalent KRAS-G12C inhibitors such as AMG-510. BI-2852 dismisses GEF, GAP, and effector interactions with KRAS, thereby causing the inhibition of downstream signaling and the proliferation in KRAS-mutant cells [75]. Interestingly, BI-2852 was identified to induce the β-β dimerization of KRAS-G12D, which may be responsible for the inhibitory activity of BI-2852 (Figure 5) [73].

### 5.4. Blocking KRAS-G12D with B-Raf Interaction

In addition to self-dimerization, the interaction of KRAS with other proteins plays an important role in the activity of KRAS. Therefore, blocking the interaction of KRAS and its activators would downregulate the downstream signaling of KRAS. Using a combination of computational and biochemical approaches, Stockwell’s group identified a site adjacent to proline 110 (P110 site). A compound KAL-21404358 was suggested to target the P110 site of KRAS-G12D, which disrupted the interaction of KRAS-G12D with B-Raf to inhibit the RAF-MEK-ERK and PI3K-AKT signaling (Figure 5) [76]. Interestingly, Shokat et al. developed a bifunctional small molecular ligand (Figure 5, Compound-12), which was able to serve as a molecular glue to promote the association between cyclophilin A and GTP-bound Ras, which disrupted the interaction with B-Raf. Such a strategy sheds light on the development of novel KRAS inhibitors [77]. Revolution Medicine developed a non-covalent pan-RAS inhibitor RMC-6236 that sticks cyclophilin A to all forms of KRAS. RMC-6236 exhibits a cancer-killing feature. However, the pan-inhibitory aspects, including affecting wild-type KRAS, induces additional safety risks [15].

### 5.5. Salt Bridge Strategy

Unlike the KRAS-G12C mutant, which could form a cysteine-based covalent bond, there was a challenge to develop selective inhibitors targeting KRAS-G12D. Considering the α-carboxylic acid moiety of Asp12 is deprotonated under physiological conditions, targeting the Asp12 residue of KRAS-G12D may produce the selectivity for KRAS-G12D. Zhang’s group replaced the acryloyl moiety group of the G12C inhibitor MRTX22 with a piperazine moiety, which was near to Asp12 to form a salt bridge interaction. As a result, both ITC and enzymatic assays indicated the selectivity of the designed inhibitor TH-Z816 for KRAS-G12D over KRAS-WT, and a salt bridge was identified through the crystal structure (PDB ID: 7EW9) (Figure 6) [78]. Further optimization of such inhibitors led to more potent compounds including TH-Z827, TH-Z835, and TH-Z837. This strategy has also been applied to the development of other KRAS-G12D inhibitors, including the MRTX1133 [11,79], which bears a piperazine moiety to improve the selectivity for KRAS-G12D. Strikingly, the protonated piperazinyl group produces 10-fold selectivity over KRAS-WT via forming a salt bridge with Asp12 (Figure 6). The piperazine adopts the twist-boat conformation, and its salt bridge to Asp12 is stabilized by an additional hydrogen bond interaction with Gly60 (Crystal structure, PDB ID:7RPZ) [11]. MRTX1133 efficiently inhibited the phosphorylation of KRAS downstream factor ERK1/2 and exhibited more than 1000-fold selectivity cell viability in KRAS-G12D-mutant cell lines compared to KRAS-WT cell lines. Meanwhile, MRTX1133 indicated dose-dependent inhibition of KRAS-mediated signal transduction and induced significant regression of the tumors (≥30%) in the patient-derived KRAS-G12D-mutant xenograft models [80].

### 5.6. Multivalent Strategy

Multivalency for ligands such as glycans may enhance both affinity and binding specificity, which has become a useful strategy in drug development [81]. For the inhibitor of KRAS, through molecular docking, multiple fragments were predicted to bind with the D38 site, A59 Site, or Y32 Site, respectively (Figure 7A). A series of multivalent compounds were designed by conjugating the fragments and were further filtered through drug-like properties prediction. The candidate compounds were synthesized and tested. The compound 3144 was identified as the most promising inhibitor (Figure 7B). Compound 3144 was identified to bind to RAS proteins as a result of microscale thermophoresis, nuclear magnetic resonance spectroscopy, and isothermal titration calorimetry, demonstrating cytotoxicity in RAS-dependent cells. It showed metabolic stability during liver microsomes assay and exhibited anti-tumor activity in vivo [82]. Therefore, the multivalent strategy is supposed to be an effective method for targets including KRAS [82].

## 6. Discussion and Perspective

Mutated KRAS plays a key role in the generation and development of various cancers, thereby becoming a promising target in anti-cancer drug development. Although KRAS has been considered an undruggable target for a long time, novel strategies have been applied to develop the directed binding inhibitors of KRAS, which include a covalent binding strategy, targeted protein degradation strategy, targeting protein and protein interaction strategy, salt bridge strategy, and multivalent strategy, etc. Huge progress has been made, and multiple potent KRAS inhibitors have been developed, such as AMG-510, MRTX849, MRTX1133, and 3144. Among the inhibitors, AMG-510 and MRTX849 have been approved by the FDA for the treatment of advanced NSCLC in patients with KRAS-G12C. Several other inhibitors, such as MRTX1133 and RMC-6236, are ready to enter clinical trials. Therefore, the multiple strategies summarized in the review may promote the development of KRAS inhibitors, which would shed light on other undruggable targets including c-myc and p53, etc. [83].

Despite the success of AMG-510 and MRTX849 in cancer with KRAS-G12C, quite a number of patients exhibit little objective response due to innate resistance [84]. This is caused by alternative carcinogenic pathways induced by the genomic heterogeneity in these cancers, such as PD-L1 expression [85] or co-mutations in STK11 and KEAP1 [86]. In such cases, biomarker assays are required to evaluate the unique factors that cause innate resistance. Moreover, the treatment of the selective KRAS inhibitors would also cause the acquired resistance, including secondary or concurrent KRAS resistance alterations, the activation of upstream, downstream, or parallel bypass signaling, the alteration of TME, and even histological transformation [87]. A recent mutagenesis screen discovered both resistant and sensitizing secondary KRAS mutations for the clinical KRAS-G12C inhibitors AMG-510 and MRTX849 [88]. Therefore, on the one hand, the combination of KRAS inhibitors with other drugs targeting the resistance-related pathway would be meaningful for better therapeutic effects. On the other hand, the KRAS-mutant-driven resistances would induce the demand for a new generation of KRAS inhibitors to conquer the drug resistance caused by the novel mutations.

In addition to small molecular inhibitors, there are multiple peptide or antibody inhibitors that directly bind to KRAS protein. Niu’s group developed a cell-permeable cyclic D-peptide NKTP-3 targeting both NRPC and KRAS-G12D using structural-based design, which showed good biostability and cellular uptake [89]. Suga’s group identified three cyclic peptide ligands of KRAS-G12D that are preferentially bound to GTP-state KRAS-G12D, thereby blocking its interaction with Raf [90]. Chen et al. found a potent peptide inhibitor memrasin, which blocked the association of KRAS with the membrane and led to the inhibition of viability of several NSCLCs in a KRAS-dependent manner [91]. Moreover, the antibodies for KRAS are attractive ways to block oncogenic KRAS signaling and provide a plausible platform for the degradation of KRAS protein. Partridge’s group built a protein-based degrader approach, which focused on the intracellular expression of a fused protein consisting of a high-affinity KRAS binding motif and an E3 ligase adapter, and led to the efficient degradation of KRAS [92,93]. Similarly, Sapkota et al. developed an “AdPROM” system for targeted protein degradation using peptidic high-affinity binders, which was applied to the downregulation of KRAS and resulted in the degradation of KRAS. Therefore, despite the deficiency of pharmacokinetic properties, macromolecules such as peptides or antibodies can provide alternative strategies to overcome KRAS oncogenic mutation [94].

Undoubtedly, mutant KRAS signaling remains the key player in anti-cancer drug development. The further strategies for targeting mutant KRAS are mostly concerned with the development strategies such as covalent binding strategy, targeted protein degradation strategy, etc. For the covalent binding strategy, more and more covalent warheads are being developed to generate various covalent bonds with multiple types of amino acids [49,95]. For targeted protein degradation strategy, besides PROTAC and molecule glue, lysosome-autophagy-based degradation techniques including AUTAC, ATTEC, AUTOTAC, or CMA-based degraders have been emerging [64]. Moreover, in addition to directly binding with KRAS, several modulators that regulate KRAS signaling are also valuable in treating KRAS-mutant-driven cancer. For example, it was reported that statin-mediated inhibition of RAS prenylation activates endoplasmic reticulum (ER) stress to enhance the immunogenicity of KRAS-mutant cancer [96].

## Figures and Tables

**Figure 1 molecules-28-03615-f001:**
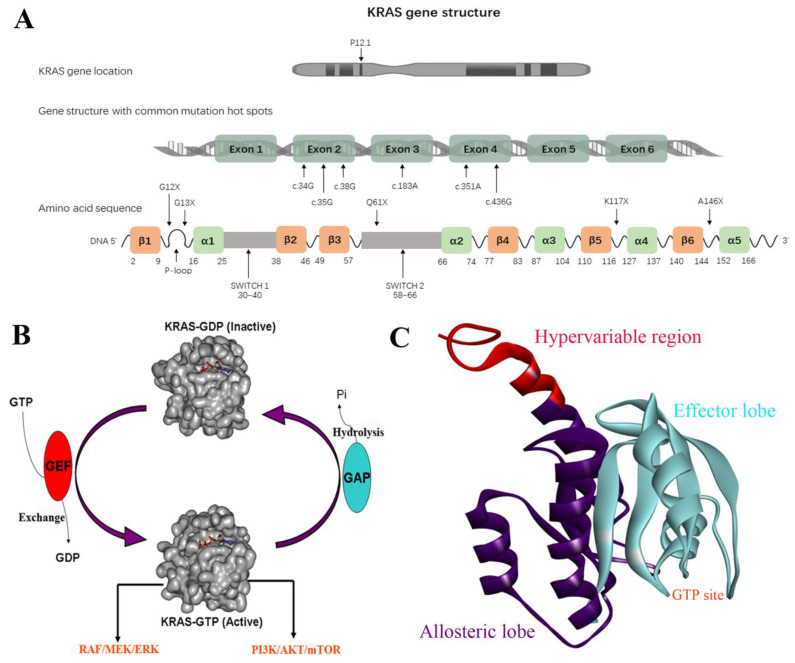
(**A**) KRAS gene structure. (**B**) The interconversion between the active GTP-bound state and the inactive GDP-bound state. (**C**) The domains of KRAS.

**Figure 2 molecules-28-03615-f002:**
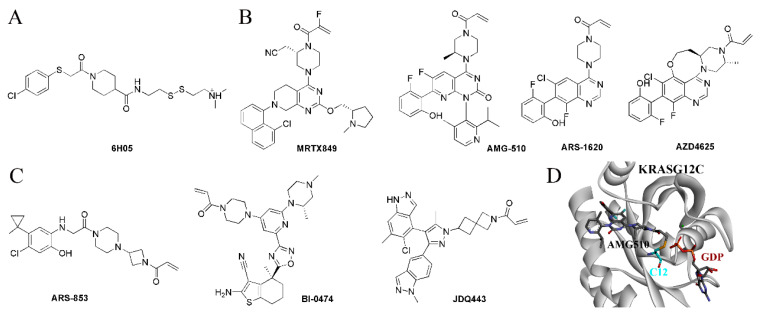
The KRAS-G12C-specific inhibitors. (**A**) The structure of 6H05. (**B**) The quinazoline ring-based inhibitors. (**C**) The inhibitors with other core structures. (**D**) The covalent bond between AMG-510 and C12.

**Figure 3 molecules-28-03615-f003:**
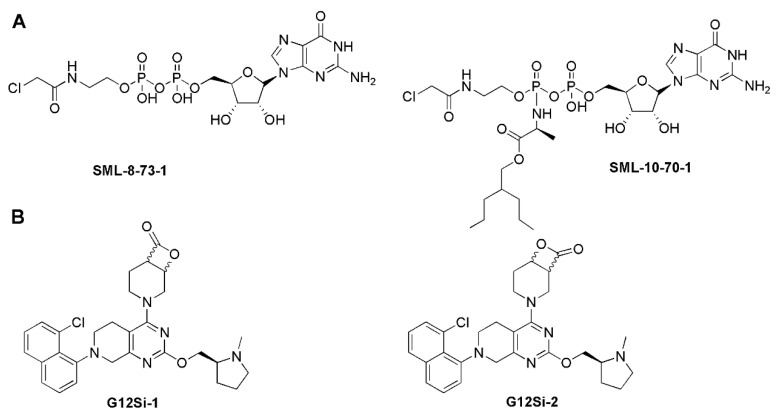
(**A**) Covalent substrate (GTP/GDP)-competitive inhibitors. (**B**) The KRAS-G12S-specific inhibitors.

**Figure 4 molecules-28-03615-f004:**
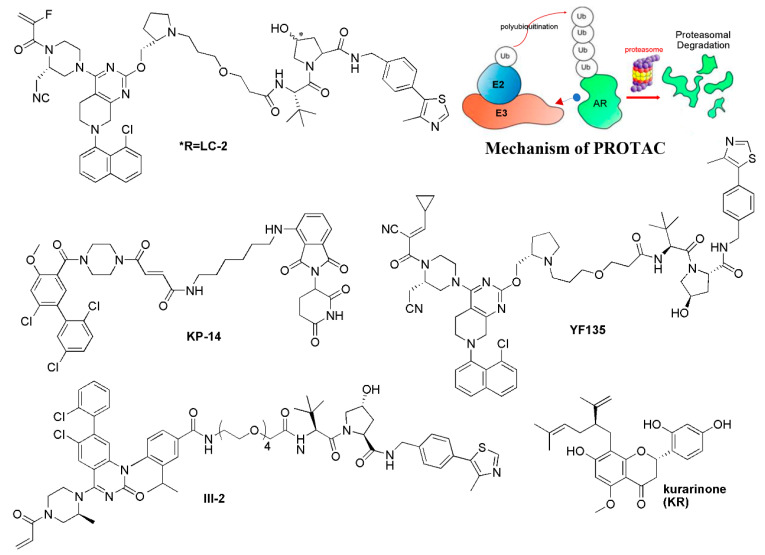
The targeted protein degradation strategy for KRAS inhibitors.

**Figure 5 molecules-28-03615-f005:**
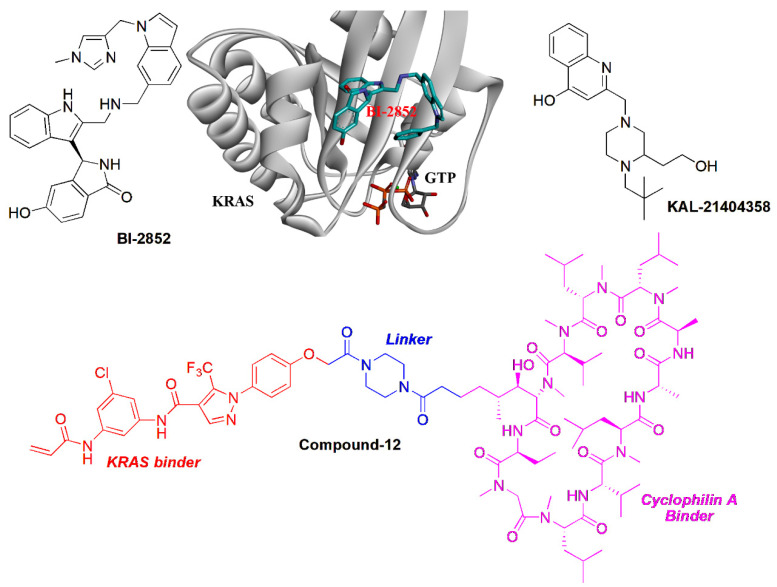
The targeting protein and protein interaction strategies for KRAS inhibitors.

**Figure 6 molecules-28-03615-f006:**
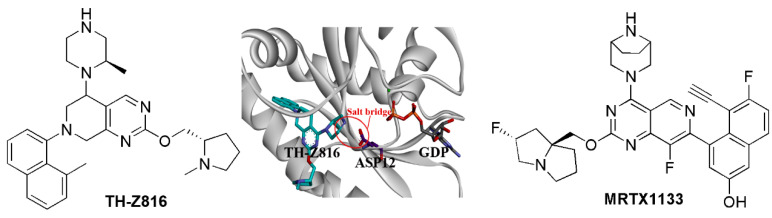
The salt bridge strategy for KRAS inhibitors.

**Figure 7 molecules-28-03615-f007:**
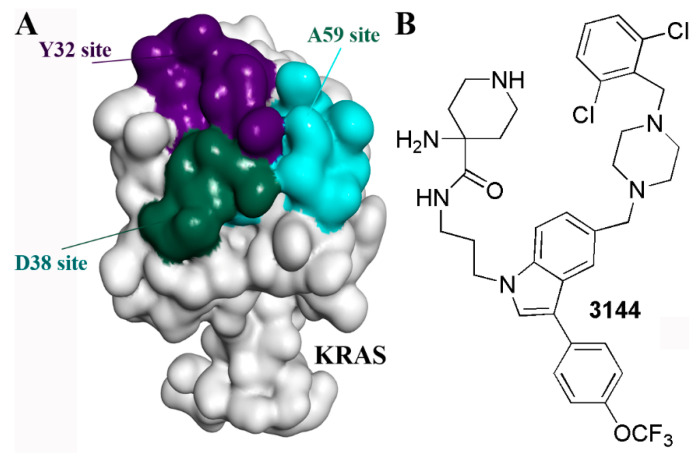
The multivalent strategy for KRAS inhibitors. (**A**) The multiple sites on the KRAS surface. (**B**) The structure of 3144.

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
