# Peer review of "Multiple Strategies to Develop Small Molecular KRAS Directly Bound Inhibitors"

_molecules, 2023, doi:10.3390/molecules28083615_

Round 1

Reviewer 1 Report

In this review, the authors evaluate new therapeutic strategies directed against KRAS. The article is both well written and organized. The authors could insert the differences between KRAS, HRAS and nRAS. They could also talk about the role of statins and regulators of RAS localization in the membrane.

They might also consider these recent articles:

  Cancer Gene Therapy volume 29, pages875–878 (2022).

Nam GH, Kwon M, Jung H, Ko E, Kim SA, Choi Y, Song SJ, Kim S, Lee Y, Kim GB, Han J, Woo J, Cho Y, Jeong C, Park SY, Roberts TM, Cho YB , Kim IS. Statin-mediated inhibition of RAS prenylation activates ER stress to enhance the immunogenicity of KRAS mutant cancer. J Immunother Cancer. 2021 Jul;9(7):e002474. doi: 10.1136/jitc-2021-002474.

Author Response

Response for Reviewer

In this review, the authors evaluate new therapeutic strategies directed against KRAS. The article is both well written and organized. The authors could insert the differences between KRAS, HRAS and nRAS. They could also talk about the role of statins and regulators of RAS localization in the membrane.

They might also consider these recent articles:

  Cancer Gene Therapy volume 29, pages875–878 (2022).

Nam GH, Kwon M, Jung H, Ko E, Kim SA, Choi Y, Song SJ, Kim S, Lee Y, Kim GB, Han J, Woo J, Cho Y, Jeong C, Park SY, Roberts TM, Cho YB , Kim IS. Statin-mediated inhibition of RAS prenylation activates ER stress to enhance the immunogenicity of KRAS mutant cancer. J Immunother Cancer. 2021 Jul;9(7):e002474. doi: 10.1136/jitc-2021-002474.

Submission Date

13 March 2023

Date of this review

17 Mar 2023 16:14:20

Response:

Many thanks. We've cited the references in the “6. Discussion and perspective”section

Reviewer 2 Report

Dear authors,

A very well structured and written review. Although, some parts need your attention.

1) Line 3 of section 1: in 1 in 7 of all human cancers, this phrase should be changed. Maybe consider “in 1 out of 7 human cancers”?

Same remark goes also to line 2 of Section 3 where the same phrase is used

2) Regarding remark 1, the same sentence is used twice but different citations are used (i.e. [1] & [31]); maybe considering citing both literatures in both cases (e.g. [1, 2] where to 2 will be the old 31)?

3) Line 16 of Section 1: exhibit should be replaced with “exhibited”

4) Line 18 of Section 1: have been should be replaced with “being”

5) Line 22 to the end of Section 1: KRAS-targeted……recent trends in KRAS inhibitors, need proper rephrasing uniformly

6) Lines 23-25 of Section 3: The mutations…..different mechanisms. Need proper rephrasing, comma is not introduced properly and messes up the sentence.

7) Line 28 of Section 3: the activation should be replaced with “the activated”

8) Line 37 of Section 3 (page 4/15): KRAS mutations….co-mutations that, needs rephrasing

9) Line 17 of Section 4: Firstly, although as beginning of the sentence is over excessive. Please rephrase accordingly

10) Line 21 of Section 4: Otherwise is not properly used here in connection with previous sentence

11) Section 5.1: The starting sentence “The covalent” until “strategy, which greatly” needs rephrasing

12) Figure 2: AMG-510 chemical structure is missing a nitrogen, the core is a pyridopyrimidinone instead of presented quinazolinone

13) Lines 13-14 of Section 5.1.: Please rephrase “Cysteine is the most nucleophilic nucleophile among the”

14) Line 22 of Section 5.1.: adagrasib (MRTX849) and sotorasib (AMG-510) are referenced first time in page 1, should you consider writing there the drugs along with their code name in brackets?

15) Figure 3: Chemical structure in figure is missing a letter “SML-10-70-1” instead of typo SM

16) Structure is also wrong, the 2-propylpentyl alaninate substitution should be in the later phosphonate and not this one!

17) Page 6 first paragraph need some attention in English

18) Page 6 second paragraph the authors intentionally wanted to say ontogenetic? Going for the entity or is it a typo?

19) As to my understanding the whole section 5.3. when it refers to “targeting protein and protein interaction” wants to address the polypharmacology approach. Stating “targeting protein and protein interaction” might be mixed with protein-protein interactions and prove misleading for some readers. Actions are required.

20) Line 3 of Section 5.4: Interestingly and considering is not combining well…

21) Figure 6: Compound from PDB entry 7ew9 is properly given in its chemical structure nut the code name is TH-Z816 instead of THZ827.

Best regards

Author Response

Response for Reviewer

A very well structured and written review. Although, some parts need your attention.

1) Line 3 of section 1: in 1 in 7 of all human cancers, this phrase should be changed. Maybe consider “in 1 out of 7 human cancers”?

Same remark goes also to line 2 of Section 3 where the same phrase is used

Response:

Thanks for the suggestion. We've rephrased “in 1 in 7 of all human cancers” as

in 1 out of 7 human cancers”.

2) Regarding remark 1, the same sentence is used twice but different citations are used (i.e. [1] & [31]); maybe considering citing both literatures in both cases (e.g. [1, 2] where to 2 will be the old 31)?

Response:

Thanks for the suggestion. We have amended the citation by replacing the [31] with [2].

3) Line 16 of Section 1: exhibit should be replaced with “exhibited”

Response:

Many thanks. We have revised it.

4) Line 18 of Section 1: have been should be replaced with “being”

Response:

Many thanks. We have revised it.

5) Line 22 to the end of Section 1: KRAS-targeted……recent trends in KRAS inhibitors, need proper rephrasing uniformly

Response:

Thanks for the suggestion. We've rephrased “KRAS-targeted……recent trends in KRAS inhibitors” as “Therefore, the drug development targeting KRAS-targeted agents provides efficient therapeutic agents for KRAS mutation-driven tumors and sheds light on the drug discovery of undruggable targets. Herein, from the view of drug development, we focus on recent trends in KRAS inhibitors”

6) Lines 23-25 of Section 3: The mutations…..different mechanisms. Need proper rephrasing, comma is not introduced properly and messes up the sentence.

Response:

Thanks for the suggestion. We've rephrased “The mutations…..different mechanisms.” as “The mutations in these codons cause an increase in GTP binding affinity and the activation of KRAS to induce downstream signaling through different mechanisms.”

7) Line 28 of Section 3: the activation should be replaced with “the activated”

Response:

Many thanks. We have revised it.

8) Line 37 of Section 3 (page 4/15): KRAS mutations….co-mutations that, needs rephrasing

Response:

Thanks for the suggestion. We've rephrased “KRAS mutations….co-mutations.” as “Moreover, KRAS mutations often combine with specific co-mutations in TP53, STK11, or KEAP1 etc., which that modulate the function of KRAS and lead to oncogenesis.”.

9) Line 17 of Section 4: Firstly, although as beginning of the sentence is over excessive. Please rephrase accordingly

Response:

Thanks for the suggestion. We've rephrased “Firstly, although” as “Despite”.

10) Line 21 of Section 4: Otherwise is not properly used here in connection with previous sentence

Response:

Thanks for the suggestion. We've rephrased “Otherwise” as “Moreover”.

11) Section 5.1: The starting sentence “The covalent” until “strategy, which greatly” needs rephrasing

Response:

Thanks for the suggestion. We've rephrased “The covalent” until “strategy, which greatly” as “As more and more covalent drugs have been successfully used in the clinic, covalent binding strategies have attracted great interest in drug development[48, 49]. For inhibitors targeting mutant KRAS, the covalent binding strategy represents a major breakthrough in drug development”.

12) Figure 2: AMG-510 chemical structure is missing a nitrogen, the core is a pyridopyrimidinone instead of presented quinazolinone

Response:

Many thanks. We have revised it.

13) Lines 13-14 of Section 5.1.: Please rephrase “Cysteine is the most nucleophilic nucleophile among the”

Response:

Many thanks. We have revised it.

14) Line 22 of Section 5.1.: adagrasib (MRTX849) and sotorasib (AMG-510) are referenced first time in page 1, should you consider writing there the drugs along with their code name in brackets?

Response:

Thanks for the suggestion. We've removed the drug names and only kept the code name here.

15) Figure 3: Chemical structure in figure is missing a letter “SML-10-70-1” instead of typo SM

Response:

Thanks for the suggestion. We've revised it.

16) Structure is also wrong, the 2-propylpentyl alaninate substitution should be in the later phosphonate and not this one!

Response:

Thanks for the suggestion. We've revised it.

17) Page 6 first paragraph need some attention in English

Response:

Thanks for the suggestion. We've revised it.

18) Page 6 second paragraph the authors intentionally wanted to say ontogenetic? Going for the entity or is it a typo?

Response:

Thanks for the suggestion. It is a typo, which should be “oncogenetic”.

19) As to my understanding the whole section 5.3. when it refers to “targeting protein and protein interaction” wants to address the polypharmacology approach. Stating “targeting protein and protein interaction” might be mixed with protein-protein interactions and prove misleading for some readers. Actions are required.

Response:

Thanks for the suggestion. The section is about to develop the blockers for self-interaction of KRAS, the interaction of KRAS with other proteins such as B-Raf. We have separated the section into two sections “Targeting the dimerization of oncogenic KRAS strategy” and “Blocking KRAS-G12D with B-Raf interaction”

20) Line 3 of Section 5.4: Interestingly and considering is not combining well…

Response:

Many thanks. We've revised it.

21) Figure 6: Compound from PDB entry 7ew9 is properly given in its chemical structure nut the code name is TH-Z816 instead of THZ827.

Response:

Many thanks. We've revised it both in Figure 6 and in text.

Reviewer 3 Report

-        Mention the KRAS abbreviation (what does KRAS stand for?) .

-       In page 3, split this sentence “For the GTP-bound on state, the γ-phosphate of GTP forms two hydrogen bonds with Thr35 (switch I) and Gly60 (switch II), respectively thereby holding the two switch regions in an active conformation to facilitate the recruitment of effector proteins”.

-       In page 4 “develop compounds that compete effectively with them for an effective blockage.”

-       In figure 2B: AMG-510, looks like the OH is bonded to the methyl group on the pyridine. Please edit.

-       In page 5: modify the sentence “which showed cell permeability and competitively inhibited KRAS-G12C”.

-       In page 6: modify the sentence “Goldsmith et al. developed the covalent inhibitor RMC-9805 (structure undisclosed)”.

-       The structure of the PROTAC YF135 is not shown.

-       The authors should discuss the future trends and their opinion about the topic in the “Discussion and perspective” part.

Author Response

Response for Reviewer

Mention the KRAS abbreviation (what does KRAS stand for?) .

Response:

Many thanks. We've revised it.

-       In page 3, split this sentence “For the GTP-bound on state, the γ-phosphate of GTP forms two hydrogen bonds with Thr35 (switch I) and Gly60 (switch II), respectively thereby holding the two switch regions in an active conformation to facilitate the recruitment of effector proteins”.

Response:

Many thanks. We've revised the sentence as “For the GTP-bound on state, the γ-phosphate of GTP forms two hydrogen bonds with Thr35 (switch I) and Gly60 (switch II), respectively, which holds the two switch regions in an active conformation to facilitate the recruitment of effector proteins”.

-       In page 4 “develop compounds that compete effectively with them for an effective blockage.”

Response:

Many thanks. We've revised it “, making it impractical to develop compounds with an effect blockage”.

-       In figure 2B: AMG-510, looks like the OH is bonded to the methyl group on the pyridine. Please edit.

Response:

Many thanks. We've revised it.

-       In page 5: modify the sentence “which showed cell permeability and competitively inhibited KRAS-G12C”.

Response:

We've revised the sentence as which improved the cell permeability and competitively inhibited KRAS-G12C through covalent binding.

-     

 In page 6: modify the sentence “Goldsmith et al. developed the covalent inhibitor RMC-9805 (structure undisclosed)”.

Response:

We've revised the sentence asGoldsmith et al. developed a covalent inhibitor RMC-9805,for which the structure is not disclosed.”

-     

 The structure of the PROTAC YF135 is not shown.

Response:

Many thanks. We've added the structure of YF135 to Figure4.

-       The authors should discuss the future trends and their opinion about the topic in the “Discussion and perspective” part.

Response:

Many thanks. We have added a new paragraph for the discussion of future trends as follows.

Undoubtedly, mutant KRAS signaling remains the key player in anti-cancer drug development. The further strategies for targeting mutant KRAS are mostly concerned with the developing strategies like covalent binding strategy, targeted protein degradation strategy, etc. For the covalent binding strategy, more and more covalent warheads are developed to generate various covalent bonds with multiple types of amino acids.[49, 95] For targeted protein degradation strategy, besides PROTAC and molecule glue, lysosome-autophagy-based degradation techniques including AUTAC, ATTEC, AUTOTAC, or CMA-based degraders have been emerging.[64] Moreover, besides directly binding with KRAS, several modulators that regulate KRAS signaling are also valuable in anti-KRAS mutant-driven cancer. For example, it was reported that statin-mediated inhibition of RAS prenylation activates endoplasmic reticulum (ER) stress to enhance the immunogenicity of KRAS mutant cancer.[96]